# Food Environments and Hepatocellular Carcinoma Incidence

**DOI:** 10.3390/ijerph18115740

**Published:** 2021-05-27

**Authors:** Mimi Ton, Michael J. Widener, Peter James, Trang VoPham

**Affiliations:** 1Department of Epidemiology, University of Washington School of Public Health, 3980 15th Ave NE, Seattle, WA 98195, USA; tvopham@fredhutch.org; 2Cancer Prevention Program, Division of Public Health Sciences, Fred Hutchinson Cancer Research Center, 1100 Fairview Ave N, Seattle, WA 98109, USA; 3Department of Geography and Planning, University of Toronto, 100 St. George Street, Toronto, ON M5S 3G3, Canada; michael.widener@utoronto.ca; 4Department of Population Medicine, Harvard Medical School and Harvard Pilgrim Health Care Institute, 401 Park Drive, Boston, MA 02215, USA; pjames@hsph.harvard.edu; 5Department of Environmental Health, Harvard T.H. Chan School of Public Health, 655 Huntington Avenue, Boston, MA 02115, USA; 6Epidemiology Program, Division of Public Health Sciences, Fred Hutchinson Cancer Research Center, 1100 Fairview Ave N, Seattle, WA 98109, USA

**Keywords:** food environments, liver cancer, epidemiology

## Abstract

Research into the potential impact of the food environment on liver cancer incidence has been limited, though there is evidence showing that specific foods and nutrients may be potential risk or preventive factors. Data on hepatocellular carcinoma (HCC) cases were obtained from the Surveillance, Epidemiology, and End Results (SEER) cancer registries. The county-level food environment was assessed using the Modified Retail Food Environment Index (mRFEI), a continuous score that measures the number of healthy and less healthy food retailers within counties. Poisson regression with robust variance estimation was used to calculate incidence rate ratios (IRRs) and 95% confidence intervals (CIs) for the association between mRFEI scores and HCC risk, adjusting for individual- and county-level factors. The county-level food environment was not associated with HCC risk after adjustment for individual-level age at diagnosis, sex, race/ethnicity, year, and SEER registry and county-level measures for health conditions, lifestyle factors, and socioeconomic status (adjusted IRR: 0.99, 95% CI: 0.96, 1.01). The county-level food environment, measured using mRFEI scores, was not associated with HCC risk.

## 1. Introduction

Hepatocellular carcinoma (HCC) comprises 85–90% of primary liver cancer cases [1,2]. HCC incidence and mortality have generally increased over time [1]. Furthermore, geographic variation in incidence and mortality suggests that environmental exposures may play a role in etiology [3]. HCC risk factors include chronic hepatitis B virus (HBV) infection, chronic hepatitis C virus (HCV) infection, aflatoxin, alcohol consumption, obesity, diabetes, metabolic syndrome, and tobacco [4]. Some research suggests that dietary factors may be associated with HCC risk. There is evidence suggesting that a lower vegetable intake [5], low selenium levels [6] and a higher intake of red meat and saturated fats [7,8] are associated with an increased risk of HCC, while inverse associations have been observed for the higher intake of plant-based fats and protein [9,10], whole grains and fiber [11], tree nuts [12], milk and yogurt, white meats, eggs, fruit, and tofu [13,14,15]. There is also evidence of dietary patterns impacting HCC development, where improved diet quality (measured by the Alternative Healthy Eating Index or AHEI) may reduce the risk of HCC [16].

However, the role of food environments in liver cancer has not been well explored. Several studies have shown that food environments (i.e., the spatial distribution of food retailers) providing more opportunities to purchase nutritious options may be important in preventing and managing chronic diseases such as obesity and cardiovascular disease, as it relates to a healthy diet [17,18,19]. Two studies showed that unhealthy food availability was positively associated with colorectal cancer incidence [20,21]. While some research has examined the association between factors related to the food environment, such as geographic variability in the dietary intake of nitrate, nitrite, and nitrosodimethylamine, and liver cancer risk [22], these studies did not specifically assess the physical food environment. To our knowledge, no epidemiologic studies have investigated the association between the food environment and HCC risk. Healthful food availability, which can be measured based on the geographic density of healthy food retailers, may impact patterns of HCC incidence [23]. 

To address geographic variability in HCC incidence and the limited research on the potential role of the food environment, we examined data from population-based cancer registries across the U.S., combining individual- and county-level data on case characteristics, health conditions, lifestyle factors, and socioeconomic status (SES). The objective of this study was to prospectively examine the association between the county-level food environment using the Modified Retail Food Environment Index (mRFEI) and HCC incidence in the U.S.

## 2. Materials and Methods

The Surveillance, Epidemiology, and End Results (SEER) database is a National Cancer Institute program, collecting individual-level information on cancer incidence from population-based cancer registries covering 34.6% of the U.S. population [24,25,26]. HCC cases identified from the following 19 registries were included in the analysis: Atlanta (metropolitan); Greater California; Connecticut; Detroit (metropolitan); Greater Georgia; Idaho; Iowa; Kentucky; Los Angeles; Louisiana (excluding July–December 2005 cases due to Hurricanes Katrina and Rita); Massachusetts; New Jersey; New Mexico; New York; Rural Georgia; San Francisco-Oakland; San Jose-Monterey; Seattle (Puget Sound); and Utah. A total of 727 counties were located in the catchment areas captured by these 19 SEER registries; these registries did not restrict coverage to specific populations and were located in the contiguous U.S., with available air pollution and ultraviolet radiation covariate data as potential confounders (a priori factors determined to be considered in modeling) [27,28]. To protect patient confidentiality, the SEER database does not include personal identifiers. This study was exempt from an Institutional Review Board review.

The inclusion criteria for cases in this study were the following: individuals residing in the area captured by the 19 SEER registries, and diagnosed with incident-confirmed HCC. HCC cases were defined using the International Classification of Diseases for Oncology, Third Edition (ICD-O-3) topography code C22.0 for primary liver cancer and ICD-O-3 histology codes 8170 to 8175 [29]; diagnostic confirmation (e.g., positive histology) excluding clinical diagnosis only; a sequence number of one primary only; diagnosis between 2000 and 2016; and not reported via autopsy or death certificate only. As conducted in previous SEER-based epidemiologic studies, the counts of HCC cases were stratified by age at diagnosis, sex, race/ethnicity, year of diagnosis, and SEER registry [30]. Each county was associated with one SEER registry.

The food environment was estimated for each county in the study area using the Modified Retail Food Environment Index (mRFEI) [31]. The mRFEI is a continuous score that measures the number of healthy and less healthy food retailers within counties, defined by the typical food offerings in specific types of retail stores (e.g., supermarkets, convenience stores, and fast-food restaurants). The number of food retailers was obtained from the U.S. Department of Agriculture (USDA) Food Environment Atlas for 2009; sensitivity analyses were performed for the number of retailers in 2014 [32]. The standard classification of healthy and less healthy food retailers was used, with grocery stores, supercenters, club stores, and farmers’ markets considered to be healthy, and fast-food restaurants, full-service restaurants, and convenience stores considered to be less healthy. The mRFEI was calculated for each county using the following formula:$$\begin{array}{c} mRFEI\\ =100\times \frac{\#\;Healthy\;Food\;Retailers}{\#\;Healthy\;Food\;Retailers+\#\;Less\;Healthy\;Food\;Retailers} \end{array}$$
$$\begin{array}{c} \#\;Healthy\;Food\;Retailers\\ =sum\left(grocery\;stores,supercenters,club\;stores,farmers^{\prime }\;markets\right) \end{array}$$
$$\begin{array}{c} \#\;Less\;Healthy\;Food\;Retailers\\ =sum\left(fast\;food\;restaurants,\;full\;service\;restaurants,\;convenience\;stores\right) \end{array}$$
mRFEI scores ranged from 0 to 100, with higher values suggesting healthier food environments. The calculated mRFEI values were linked to each county via U.S. Federal Information Processing Standard (FIPS) codes, using the county of residence at diagnosis that was available for each case from the SEER registries. Spatial analyses were conducted in ArcGIS version 10.8 (Esri, Redlands, CA, USA).

Individual- and county-level information on established or suspected HCC risk factors and/or variables known to be associated with the exposure were evaluated as potential confounders. From the SEER database, we acquired individual-level data on age at diagnosis (years), sex (male, female), race/ethnicity (non-Hispanic white, non-Hispanic black, Hispanic, non-Hispanic Asian or Pacific Islander, non-Hispanic American Indian or Alaska Native), year of diagnosis (2000–2016), and SEER registry. From the 2000 U.S. Census Bureau summary files available through SEER, we obtained county-level data on median household income (per $10,000), poverty (percentage below the poverty level), percentage unemployed, educational attainment (percentage with a bachelor’s degree or higher), and the percentage of foreign-born individuals. Percentage of foreign-born data were used as a proxy for HBV prevalence, which is endemic in parts of Asia and Africa [33]. We also acquired information on county-level urbanicity (rural, urban) using USDA Rural-Urban Continuum Codes, which was used as a proxy for HCV prevalence, as differences in HCV have been observed in rural vs. urban areas [34,35].

Additional county-level data were acquired from the Institute for Health Metrics and Evaluation, created using information from the U.S. Behavioral Risk Factor Surveillance System (BRFSS) and the U.S. National Health and Nutrition Examination Survey (NHANES). Information on the sex-specific age-adjusted prevalence of alcohol consumption in 2005 (average > 1 drink per day for women, or >2 drinks per day for men in the past 30 days) [36], current smoking in 2000 (currently smoking cigarettes daily or non-daily) [37], diabetes in 2000 (percentage of adults ≥ 20 years who reported a previous diabetes diagnosis and/or have a fasting plasma glucose of ≥126 mg/dL and/or hemoglobin A1c ≥ 6.5%) [38,39], and obesity in 2001 (body mass index (BMI) ≥ 30 kg/m^2^) [40] was ascertained. Ambient particulate matter < 2.5 microns in diameter (PM_2.5_) exposure (μg/m^3^) was estimated for each county using a spatial PM_2.5_ exposure model [27]. Additionally, ambient ultraviolet radiation (UV) exposure (mW/m^2^), a proxy for circulating levels of vitamin D, was estimated using a spatiotemporal geostatistical exposure model [28]. The region of residence was defined as counties located in the U.S. Census Bureau-defined regions: Northeast, Midwest, South, and West. All county-level data were compiled using unique U.S. FIPS codes.

Poisson regression with robust variance estimation was used to calculate basic and fully adjusted incidence rate ratios (IRRs) and 95% confidence intervals (CIs) for the association between the county-level food environment (using the mRFEI) and HCC risk. mRFEI scores were examined continuously per interquartile range (IQR) increase (IQR: 9.3) and categorically using quintiles estimated across all counties in the study area. All models were a priori-determined to include individual-level age at diagnosis, sex, race/ethnicity, year of diagnosis, SEER registry, and the following county-level variables: urbanicity, median household income, percentage with a bachelor’s degree or higher, percentage unemployed, percentage of individuals below the poverty level, percentage of foreign-born individuals, and the prevalence of alcohol consumption, smoking, obesity, and diabetes. We evaluated potential confounding by ambient PM_2.5_ and UV exposure, which have been shown to have positive and inverse associations with HCC risk, respectively [27,28]. After adjustment for these variables, the IRR for mRFEI and HCC risk did not substantially change and were thus not included in the final fully adjusted model. The natural logarithm of the county population was used as the offset in all models. County-level population data were acquired from the National Center for Health Statistics Bridged-Race Resident Population Estimates online database, downloaded from the Centers for Disease Control and Prevention WONDER (Wide-ranging Online Data for Epidemiologic Research) [41].

Stratified analyses were conducted to explore potential effect modification by factors that may be associated with the food environment or with disparities in HCC incidence. We examined potential interactions with age at diagnosis, sex, race/ethnicity, year of diagnosis, region, state, median household income, poverty, unemployment, educational attainment, urbanicity, alcohol consumption, smoking, diabetes, obesity, and foreign-born. Tests for interaction were conducted by adding an interaction term to the model and using likelihood ratio tests to determine statistical significance. We performed sensitivity analyses using mRFEI scores calculated using the number of food retailers in 2014, and using log-transformed mRFEI scores to account for the relatively diminishing beneficial effects of higher mRFEI values compared to lower values [42]. A *p*-value < 0.05 was considered statistically significant. All statistical analyses were performed using SAS version 9.4 (Cary, NC, USA). 

## 3. Results

A total of 90,578 HCC cases, diagnosed between 2000 and 2016, were included in the analysis. HCC cases were on average 62.7 years of age, mostly male (77.5%), non-Hispanic white (50.0%), and/or resided in the Western region of the U.S. (46.1%) (Table 1). Using county-level data from the underlying population from which HCC cases were sampled, HCC cases at the time of diagnosis resided in counties where the average mRFEI scores were 16.9 ± 8.3. HCC cases lived in counties where an average of 8.4% of the population consumed alcohol, 29.0% smoked cigarettes, 28.4% were obese, and 12.2% had diabetes. HCC cases resided in counties with a median household income of $36,300, where an average of 16.9% had a bachelor’s degree or higher, 6.3% were unemployed, 15.1% were living below the poverty level, and 4.6% were foreign-born. 

Figure 1 shows mRFEI scores in 2009, categorized by quintiles, calculated using the 727 counties included in the study. The mRFEI values ranged between 4.55 and 50. In general, higher mRFEI scores were observed in the Midwest (SEER registries in Detroit and Iowa) and lower mRFEI scores were observed in the Northeast (SEER registries in Connecticut, Massachusetts, New Jersey, and New York) regions of the U.S.

mRFEI scores were not associated with HCC risk in basic models, adjusted for age at diagnosis, sex, race/ethnicity, year of diagnosis, and SEER registry (adjusted IRR per IQR increase: 1.00, 95% CI: 0.97, 1.03, *p* = 0.97) (Table 2). After additional adjustment by county-level urbanicity, median household income, percentage with a bachelor’s degree or higher, percentage unemployed, percentage of individuals below the poverty level, percentage of foreign-born individuals, and prevalence of alcohol consumption, smoking, obesity, and diabetes, we observed no association between mRFEI scores and HCC risk (adjusted IRR per IQR increase: 0.99, 95% CI: 0.96, 1.01, *p* = 0.36). Similar null results were observed when examining quintiles of mRFEI scores.

We observed statistically significant interactions (*p* < 0.05) between mRFEI scores and age at diagnosis, race/ethnicity, year of diagnosis, and region (Supplemental Appendix A). However, the effect estimates for mRFEI scores across strata defined by these variables were similar. The association between mRFEI scores and HCC risk did not differ by sex, state, median household income, poverty, unemployment, educational attainment, urbanicity, alcohol consumption, smoking, diabetes, obesity, or foreign-born (*p* > 0.05). In sensitivity analyses, we observed similar results when examining mRFEI estimated in 2014 and using log-transformed mRFEI values (results not shown).

## 4. Discussion

This study showed no association between the county-level food environment (estimated using the mRFEI) and HCC incidence, after adjusting for individual- and county-level information on demographics, health conditions, lifestyle factors, and SES. To our knowledge, this is the first study examining the association between the food environment and HCC risk in the U.S.

Food environments may represent an important context in which cancer risk is shaped, informing disease processes and enabling the identification of vulnerable populations [43]. The food environment can include both health-promoting resources (e.g., healthy food retailers) and less healthy amenities (e.g., fast food restaurants), and has been shown to influence healthier dietary attitudes and behaviors [19,44,45,46,47]. For example, one study found that higher mRFEI scores (indicating a healthier food environment) were associated with increased levels of objectively assessed fruit and vegetable consumption [48]. Another study showed that perceptions about the food environment are important in promoting self-efficacy for consuming healthier foods, such as fruits and vegetables, among U.S. adults [49]. Access to healthier foods and healthier dietary patterns may affect cancer risk through mechanisms related to reducing chronic inflammation [50]. Population-based research on the impact of the food environment on cancer risk has not been well established, although two studies found some evidence of a positive association between an unhealthy food environment and colorectal cancer [20,21]. For HCC in particular, several studies have suggested that diet alteration can affect HCC risk. For example, the consumption of nutrients derived from plants [9,10,11,12] and dairy and proteins, such as white meat and eggs [13,14,15], may play a role in reducing HCC risk. When examining dietary patterns, adherence to the AHEI has been inversely associated with HCC risk [16]. 

In this study using SEER data, we observed no association between the county-level food environment and HCC risk after adjustment. Established risk factors for HCC have shown geographic variation across the U.S., including age [51,52], sex [52], race/ethnicity [53], obesity [54], diabetes [55], alcohol consumption [56], HBV, and HCV [57]. The median age of the population varies by state, with higher median ages (40 years or older) observed in parts of the Northeast and South [52]. Sex ratios, or the number of males per 100 females, were higher in the West and lower in the Northeast [52]. Race/ethnicity also differed geographically, with California, Texas, Washington, D.C., Hawaii, and New Mexico being characterized by a relatively higher population of race/ethnicities other than non-Hispanic white [53,58]. Obesity and diabetes also varied by state, with a higher diabetes prevalence and average BMI in parts of the South such as Mississippi (mean 30.5 kg/m^2^) [54,55]. Alcohol consumption was highest in areas including New England, and lower in parts of the South [56]. HCV also exhibits geographic variation, as the following nine states included 51.9% of all individuals living with HCV in the U.S.: California, Texas, Florida, New York, Pennsylvania, Ohio, Michigan, Tennessee, and North Carolina [57]. As geographic variability in HCC incidence across the U.S. may be explained by geographic differences in the underlying prevalence of risk factors [59], we adjusted for many of these variables at the individual or county level in our analyses. However, future research including information on individual-level HCC risk factors would be valuable.

Although there were statistically significant interactions with age at diagnosis, race/ethnicity, year at diagnosis, and region, stratified effect estimates were similar and not meaningfully different. Several factors may have contributed to our observing overall null results. The usage of administrative boundaries, such as counties, may be limited when representing the food environment. When asked to define neighborhood boundaries, residents created maps that resulted in differing spaces (30% smaller in size) than those determined by census-defined boundaries [60,61]. Thus, county-level data may not accurately represent the food environment to which individuals are exposed. Furthermore, food availability may be modified by factors such as urbanicity and transportation. For example, car ownership may expand the boundaries of the local food environment. One study found that among non-car owners, those who resided in a food environment with a higher concentration of fast-food outlets had higher BMIs compared to those who resided in an environment with relatively fewer fast-food outlets [62]. Future research could consider examining more granular locational and food environment data and other relevant covariate factors such as transportation, time-use, and household composition. In addition, our study did not account for pesticide use, a risk factor for HCC, which may be related to the food environment such as through the amount of pesticides used in fresh produce at the food retailers [63,64]. 

The limitations of our study include its ecological design, preventing inference to the individual-level risk of developing HCC. County administrative boundaries may not accurately represent the individual-level residential environment [65,66]. The usage of more granular data, such as census tracts, would increase exposure variability compared to county-level analyses of mRFEI scores [67]. Although the mRFEI score is an established measure for the food environment [23,48,68], there may be potential exposure misclassification, as there was no information on the specific food options provided at each food retailer. Furthermore, the binary grouping of establishments into healthy vs. unhealthy may not accurately represent the food environment. Alternative measures of the food environment, such as quick-service food retail availability, may capture other aspects of the food environment not considered in the mRFEI. There may also be residual confounding due to unmeasured factors. Although the prevalence of HCV in the U.S. is low (1.0–1.9%), and HCV infection may be more common in urban areas, urbanicity as a proxy for HCV infection may not adequately capture geographic variation in HCV [34,69]. Future research should include information on individual-level HCV status. There are also regional differences in the quality of food environments across the U.S., such as a lack of access to supermarkets in areas including the Great Plains, Deep South, and Appalachian region [70]. Although we addressed these regional differences through stratified analyses by region and state, additional research using locational data that are more granular than the county level is warranted. The strengths of our study include a large sample size of confirmed HCC cases from population-based cancer registries across the U.S. The study area included counties across the contiguous U.S., characterized by a wide range of mRFEI scores. Data on many potential confounders and effect modifiers at the individual and county levels were available. 

## 5. Conclusions

In conclusion, our results showed no association between the food environment and HCC risk. Future research examining the association of the food environment and HCC risk could examine higher resolution locational and food environment data, and account for transportation and dietary preferences.

## Figures and Tables

**Figure 1 ijerph-18-05740-f001:**
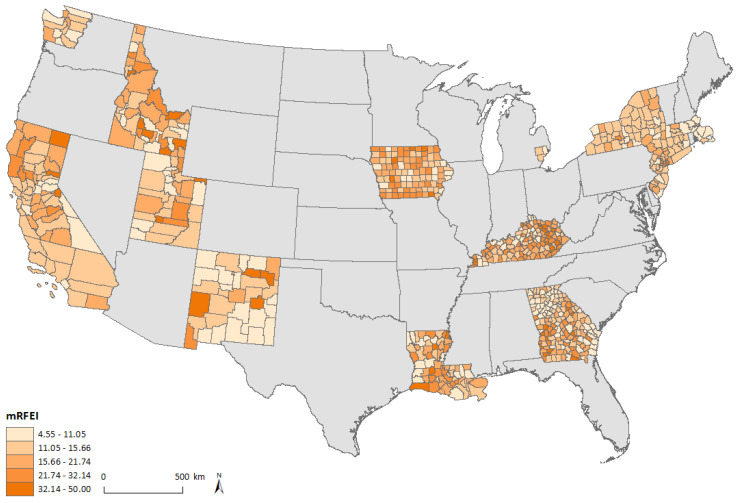
Map of mRFEI in SEER catchment areas of the contiguous United States.

**Table 1 ijerph-18-05740-t001:** Population characteristics of hepatocellular carcinoma cases in the U.S. (SEER 2000–2016).

Characteristic	*N* (%)
**Individual-level**	90,578
Age at diagnosis (years) (mean ± SD)	62.7 ± 11.5
Sex	
Male	70,167 (77.5)
Female	20,411 (22.5)
Race/ethnicity	
Non-Hispanic white	45,404 (50.0)
Non-Hispanic black	12,841 (14.2)
Hispanic	17,804 (19.7)
Non-Hispanic Asian or Pacific Islander	13,713 (15.1)
Non-Hispanic American Indian or Alaska Native	816 (0.9)
Year of diagnosis	
2000–2007	32,362 (35.7)
2008–2016	58,216 (64.3)
**County-level**	727
mRFEI ^†^ (mean ± SD)	16.9 ± 8.3
Percent alcohol consumption (mean ± SD)	8.4 ± 2.4
Percent smoking (mean ± SD)	29.0 ± 4.9
Percent obese (mean ± SD)	28.4 ± 3.6
Percent diabetes (mean ± SD)	12.2 ± 1.6
Median household income ($10,000) (mean ± SD)	36.3 ± 10.3
Percent bachelor’s degree or higher (mean ± SD)	16.9 ± 8.7
Percent unemployed (mean ± SD)	6.3 ± 2.6
Percent poverty (mean ± SD)	15.1 ± 7.2
Percent foreign-born (mean ± SD)	4.6 ± 6.4
Urbanicity	
Urban	614 (84.5)
Rural	113 (15.5)
Region	
Northeast	31,628 (34.9)
South	12,387 (13.7)
Midwest	4835 (5.3)
West	41,728 (46.1)

^† ^Modified Retail Food Environment Index.

**Table 2 ijerph-18-05740-t002:** Associations between mRFEI exposure and HCC incidence (SEER 2000–2016).

mRFEI ^†^ Exposure	N Cases	Basic ^‡^		Fully Adjusted ^§^	
		IRR (95% CI)	*p*	IRR (95% CI)	*p*
mRFEI (per IQR increase) ^¶^	90,578	1.00 (0.97, 1.03)	0.97	0.99 (0.96, 1.01)	0.36
mRFEI quintiles			0.99		0.69
Quintile 1 (<10.67)	18,857	Referent		Referent	
Quintile 2 (10.67–12.99)	38,519	1.04 (0.97, 1.12)		1.00 (0.95, 1.05)	
Quintile 3 (>12.99–16.67)	18,088	1.04 (0.96, 1.13)		1.01 (0.96, 1.07)	
Quintile 4 (>16.67–22.22)	5125	0.94 (0.87, 1.02)		0.97 (0.90, 1.04)	
Quintile 5 (>22.22)	9989	1.03 (0.93, 1.14)		0.98 (0.87, 1.10)	

^† ^Modified Retail Food Environment Index. ^‡ ^Adjusted for age at diagnosis, sex, race/ethnicity, year, and SEER registry. ^§ ^Additionally adjusted for county-level measures of urbanicity, median household income, percentage with a bachelor’s degree or higher, percentage unemployed, percentage of individuals below the poverty level, percentage of foreign-born, and prevalence of alcohol consumption, smoking, obesity, and diabetes. ^¶ ^Interquartile range, IQR: 9.3.

## Data Availability

Restrictions apply to the availability of this data. Data were obtained from SEER registries and are available with the permission of SEER.

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
