# Peer review of "Food Environments and Hepatocellular Carcinoma Incidence"

_ijerph, 2021, doi:10.3390/ijerph18115740_

Round 1
Reviewer 1 Report
The mRFEI calculation is still quite unclear, and this is a major issue. In some places (the Methods), the text says high mRFEI indicates a more healthful food environment. But in the Results, the text says "higher mRFEI was not associated with risk of HCC." Why would a healthier food environment be associated with HCC risk? This is a counterintuitive state of affairs. Also, in the map, I don't understand how the more rural parts of Georgia and Kentucky, for example, have more healthful food environments than urban areas of those same states. Something is fishy here. Please check again the mRFEI calculation and all text regarding it to clarify.
Author Response
Thank you for your comments. Please see attachment for our response.

Reviewer 2 Report
This well-written manuscript is an analysis of the relationship between food environment and hepatocellular carcinoma incidence using 2000 and 2016 SEER data. This analysis is important because the role of food environments in cancer prevention has been underexplored. The methods are clearly described, the mRFEI formula has been clarified, and the discussion has been strengthened.
Author Response

(The authors gave the same response as above.)

Round 2
Reviewer 1 Report
Added comments re: county-level vs. census tract-level mRFEI are a welcome addition to the manuscript. While I am still surprised by the county-level mRFEI patterns, I think it is publishable now.
This manuscript is a resubmission of an earlier submission. The following is a list of the peer review reports and author responses from that submission.
Round 1
Reviewer 1 Report
Abstract
Suggest striking "per interquartile range (IQR) (9.3) increase" from abstract to prioritize clarity, rather than detail here.
Intro
Re: studies of food environment and liver cancer--I did find this one article in a 2-minute GoogleScholar search: (2008) Geographic Distribution of Liver and Stomach Cancers in Thailand in Relation to Estimated Dietary Intake of Nitrate, Nitrite, and Nitrosodimethylamine, Nutrition and Cancer, 60:2, 196-203, DOI: 10.1080/01635580701649636.
It's quite different, but enough to question the authors' absolute statement re: the literature.
Methods
Consider adding a figure with additional details (types of retailers considered more/less healthy, e.g.) to show how the mRFEI was formulated. The existing formula does not really add much.
It seems odd that region is an individual-level variable, rather than a county-level variable, since it is contextual, and not portable or inherent to the individual. Rather like urbanity, in fact. Not sure how much difference this makes in the statistical wash, but conceptually it leaves a funny taste.
Results
Table 1 is difficult to read due to center-justification of both columns. Suggest left-justification of text with indention of category labels for each variable, and right-justification of numeric columns.
"In general, higher mRFEI scores were observed in the Midwest"--Do the authors refer to Iowa? That's really the only Midwestern state, other than Detroit (which isn't very high, actually). What about some other regions known for poor food environments--e.g., Eastern/Appalachian Kentucky, Deep South, reservations?! More knowledgable description would be useful here.
In Table 2, suggest replacing "Basic" with "Unadjusted", and "Fully adjusted"
Suppl. Table 1 is also difficult to read--see notes for Table 1, above.
What about other factors known to be associated with HCC? Tables don't contain any information about unadjusted or adjusted results for covariates. If the authors aren't finding any signal with known risk factors, maybe analytic design is fundamentally flawed.
Discussion
A bit anemic, overall. There is not nearly enough discussion of hepatitis C. And controlling for its influence with a rural-urban code proxy is likely wholly inadequate. This should at least be mentioned as an important limitation.
Conclusions are fine; there's just not much here. Suggest creating more interest by giving the reader a glimpse of more established risk factors and how they relate to spatial distribution of HCC. Modeling needs work, too, so additional work there may guide authors further.
Reviewer 2 Report
What foods and nutrition are risk factors and preventive factors for chronic diseases is an important research topic. Although HBV infection, HCV infection, aflatoxin, alcohol, and tobacco are known to be risk factors for hepatocellular carcinoma, which accounts for 85-90% of primary liver cancers, the role of the food environment in cancer prevention is still unknown. This paper is significant because of its practical and academic content related to the food environment and the development of hepatocellular carcinoma. The methods used were surveillance, epidemiology, and the number of cancer registries. The statistical analysis of the large number of hepatocellular carcinoma cases diagnosed between 2000 and 2016 (90578 cases) is valuable data. It is worth noting that this is the first full-scale epidemiological study on the relationship between the food environment and the development of hepatocellular carcinoma. The value of this study also lies in the use of multi-level data from population-based cancer registries across the US and the fact that it is a prospective epidemiological study. County-level food environment did not correlate with the risk of developing hepatocellular carcinoma. However, can this study exclude the effects (risk factors) of pesticides used in the growth of fresh vegetables and additives used in preserved foods? This is a matter of concern. I would like to see some additional statements in the discussion.